# Application of Extracellular Vesicles in Gynecologic Cancer Treatment

**DOI:** 10.3390/bioengineering9120740

**Published:** 2022-11-29

**Authors:** Renwen Zhang, Yixing Zou, Jing Luo

**Affiliations:** Institute of Reproductive Health, Center for Reproductive Medicine, Tongji Medical College, Huazhong University of Science and Technology, Wuhan 430030, China

**Keywords:** extracellular vesicles, exosomes, ovarian cancer, cervical cancer, endometrial cancer, function

## Abstract

Ovarian, cervical, and endometrial cancer are the three most common gynecological malignancies that seriously threaten women’s health. With the development of molecular biology technology, immunotherapy and targeted therapy for gynecologic tumors are being carried out in clinical treatment. Extracellular vesicles are nanosized; they exist in various body fluids and play an essential role in intercellular communication and in the regulation of various biological process. Several studies have shown that extracellular vesicles are important targets in gynecologic cancer treatment as they promote tumor growth, progression, angiogenesis, metastasis, chemoresistance, and immune system escape. This article reviews the progress of research into extracellular vesicles in common gynecologic tumors and discusses the role of extracellular vesicles in gynecologic tumor treatment.

## 1. Introduction

The three most common gynecologic malignancies, cervical cancer (CC), endometrial cancer (EC), and ovarian cancer (OC), seriously threaten women’s health [1]. According to cancer statistics in 2021, CC was the fourth most common cancer and the fourth leading cause of women dying from cancer with about 604,100 new cases and 341,831 deaths worldwide in 2020 [2]. EC is one of the most common gynecologic cancers in countries with high human development indexes. Its morbidity and mortality are increasing globally, with about 417,367 new cases and 97,370 deaths worldwide in 2020 [1,2,3]. OC is one of the most lethal gynecologic malignancies, with about 313,959 new cases and 207,252 deaths worldwide in 2020. Compared with healthy cells, tumor cells release more EVs and have different types of physicochemical properties [4]. With the development of gynecologic oncology, EVs are receiving more attention in gynecologic tumor treatment [5].

Extracellular vesicles(EVs) are lipid vesicles that can be secreted by most cells. The outer layer of an extracellular vesicle is wrapped by a layer of lipid membrane; its inner layer contains different types of nucleic acids and proteins, and it has a biologically active function in transmitting information between cells [6,7,8]. EVs are a collective term for multiple subtypes, which exist in body fluids such as blood, semen, urine, saliva, breast milk, amniotic fluid, and ascites [9,10]. According to the nomenclature of the International Society for Extracellular Vesicles, researchers should consider using operational terms for EV subtypes that refer to the physical characteristics of the EVs, their biochemical composition, and descriptions of the conditions or cell of origin, in place of terms such as exosome and microvesicle that have contradictory definitions and inaccurate biosignatures [11]. EVs, which are composed of phospholipids, were first identified in platelets in 1967 and were considered as waste products of cells [12]. In the following decades, an increasing number of studies found a correlation between the active components carried by exosomes secreted in the pathological setting and the development and prognosis of the pathological state, which can be applied to the detection, prognosis, and guidance of the treatment of early cancers [13,14,15]. Techniques for the detection and purification of extracellular vesicles are also developing and improving. Therefore, in this paper, we review the research progress into EVs in common gynecologic tumors and discuss the research prospects and directions of EVs in gynecologic tumor treatment (Table 1).

## 2. Progress of the Research into EVs in the Diagnosis and Treatment of OC

OC is a common malignant tumor in gynecology. It has been reported that epithelial OC is the most common type of OC, and because 60 percent of cases are diagnosed at stage III or IV, it is associated with a poor prognosis [101]. The 5-year survival rate of stage I OC can reach 89%, and it decreases to 20% after the disease develops to stage IV [101]. Due to the lack of typical symptoms and specific biomarkers, most patients with OC are not diagnosed until the advanced stages, making it the gynecologic cancer with the highest mortality [102,103]. Although several novel drugs have been developed and applied in clinical trials, the clinical cure rate for patients with advanced OC has not been significantly improved [104,105,106]. Therefore, improving the diagnostic rate of early stage disease and clinical treatment effect is crucial to reducing the morbidity and mortality of OC. Moreover, exosomes derived from OC have the potential to become new biomarkers and therapeutic targets [107].

### 2.1. OC Development and Metastasis

EVs play a vital role in the development and metastasis of OC. OC cells release exosomes into the surrounding environment, increasing the intercellular interaction with the cells related to tumor development, metastasis, and invasion. Currently, known metastatic pathways include hematogenous metastasis, lymphatic metastasis, and transperitoneal metastasis [108,109,110]. The major pathways spread to the greater omentum through the peritoneal cavity, and OC cells are shed and accumulate in the peritoneal fluid [111]. Many biomolecules have been identified in the EVs secreted by OC. These molecules affect cell signaling and alter the tumor microenvironment by inducing tumor growth and metastasis [112]. Proteomic analysis of OC-derived EVs has shown that many proteins common to EVs from various origins, including TSG101, Alix, heat shock proteins, tetraspanins, rabs, annexins, and cytoskeletal proteins, may be associated with the biogenesis, structure, and trafficking of EVs [113]. Proteins (ATF2, MTA1, ROCK1/2, and sE-cad) [16,17], long noncoding RNAs (MALAT1, lncRNA ATB) [48,49], microRNAs (miR-130a, miR-205, and miR-141-3p) [26,27,28], and a protein receptor (PKR1) [25] in EVs promote tumor angiogenesis, increase the vascular permeability, and provide a nutrient supply for tumor growth and metastasis. Arginase 1 (ARG-1) and phosphatidylserine (PS) released by EVs inhibited T cell activation and proliferation and promoted OC growth in the form of immunosuppression [50,51]. Tumor-derived extracellular vesicular miR-940 induced M2-type polarization in macrophages [29], and the M2-type macrophage extracellular vesicular miR-221-3p inhibited cyclin-dependent kinase inhibitor 1B and promoted OC proliferation and metastasis [30]. High metastatic OC cells conferred high metastatic properties to low metastatic cancer cells via extracellular vesicular circRNA051239, leading to the enhanced proliferation, migration, and invasion of the recipient cells [45]. Ascites-derived EVs transferred miR-6780b-5p to OC cells, which was shown to promote the onset of epithelial mesenchymal transition and accelerate OC metastasis [31]. Extracellular vesicular miR-21-5p inhibited the expression of cyclin-dependent kinase 6 and increased the volume, size, and weight of OC in vivo [32].

These findings demonstrate the role of EVs in promoting tumor development and metastasis and as potential targets for subsequent treatment.

### 2.2. Diagnosis and Prognosis of OC

Early precise diagnosis and late dynamic follow-up of gynecologic tumors have always been important topics. Through fluid sampling, liquid biopsy is significant for disease diagnosis, prognosis, and efficacy assessment. EVs secreted by tumor cells are available in various body fluids and can be distinguished from non-cancerous EVs, leading to their potential to be cancer biomarkers and becoming a part of the standard test for liquid biopsies [114]. In addition, the stability of EVs gives them a higher specificity and sensitivity [115].

A study first reported the diagnostic value of extracellular vesicular miRNAs in OC in 2008, identifying the upregulation of eight specific miRNAs in the serum EVs of OC patients; the study found that miR-200a/b/c was an effective indicator to identify benign and malignant ovarian tumors [33]. Research has shown that the claudin-4 protein is released from OC cells via EVs, and the claudin-4 protein obtained from peripheral blood EVs had a sensitivity of 51% and specificity of 98% differentiating between healthy people and OC patients [18]. In addition, a clinical study sample showed patients with OC had a higher expression of plasma EVs miR-21, miR-100, miR-200b, miR-320, and miR-1290 and a lower expression of miR-16, miR-93, miR-126, and miR-223 [34,35]. Similar studies showed that miR-1260a, miR-7977, and miR-192-5p expression in plasma EVs was significantly reduced in OC patients and could be used as potential diagnostic and prognostic biomarkers [36]. Moreover, researchers utilizing a microfluidic device discovered that HGF, STAT3, and IL-6 were highly elevated in the serum EVs of patients with early stage high-grade serous OC, compared to the benign and late-stage HGSOC [19]. In ovarian cancer-associated fibroblasts (CAFs)-derived EVs, transforming growth factor beta (TGFβ) was upregulated compared to the normal omentum fibroblasts [20]. The extracellular vesicular miR-21-5p, miR-29a-3p, and miR-30d-5p were significantly overexpressed in clear cell carcinoma of the ovary compared with normal cells [37].

In addition, EVs have the potential to be prognostic predictors of OC. It has been shown that the levels of miR-200a, miR-200b, miR-200c, and miR-1290 were increased in the ascites of OC patients, and the level of miR-200b was associated with overall survival rate [38]. As an EV-associated gene, the expression of fibroblast growth factor 9 (FGF9) in ovarian epithelial tissue was lower than that in normal ovarian tissue, and the downregulated FGF9 showed good prognostic value in patients with OC [21]. The plasma EV-derived fragile site-associated tumor suppressor (FATS) was significantly decreased in OC patients; meanwhile, low levels of plasma EVs-derived FATS were closely associated with the Federation International of Gynecology and Obstetrics (FIGO) stage III/IV, high grade, ascites, elevated CA-125, and lymph node metastasis and prognosis of OC patients [22]. The circRNA circular forkhead box protein P1 (circFoxp1) in the serum EVs of patients with epithelial ovarian cancer (EOC) was significantly increased, particularly in those who presented with cisplatin resistance. CircFoxp1 expression was positively correlated with the FIGO stage, primary tumor size, lymphatic metastasis, distant metastasis, residual tumor diameter, and clinical response in OC, and was an independent predictor of survival and disease recurrence for patients with EOC [46]. The expression of circRNA Cdr1a in serum EVs derived from OC patients was less expressed in the cisplatin-resistant group than in the cisplatin-sensitive group, and Cdr1as enhanced the cisplatin chemosensitivity of OC in vivo. Therefore, extracellular vesicular Cdr1a in serum can serve as a promising biomarker for cisplatin-resistant OC patients [47]. The OC patients with simultaneously low serum extracellular vesicular miR-484 expression and high serum CA-125 levels tended to suffer the worst clinical outcomes. The multivariate analysis confirmed that the low serum extracellular vesicular miR-484 level was an independent indicator [39]. Serum extracellular vesicular metastasis-associated lung adenocarcinoma transcript 1 (MALAT1) expression was higher in OC patients than in the healthy control group. High levels of extracellular vesicular MALAT1 were associated with an advanced FIGO stage, a high histological grade, and lymph node metastasis, indicating that serum extracellular vesicular MALAT1 could be used as a biomarker in the prognosis prediction of OC [48].

These results suggest that the active substances carried by EVs have the potential to become biomarkers for the diagnosis and prognosis of OC.

### 2.3. Treatment of OC

Surgery and chemotherapy are the most commonly used methods for the treatment of OC. Conventional targeted drug-controlled systems for oncology treatment cause serious side effects, including organ toxicity and the attenuation of host immune response. It is easy to develop drug resistance in the late stage of chemotherapy, which adversely affects the treatment [116,117]. Based on the properties of EVs, their clinical value can be developed by regulating their secretion to strengthen antitumor immunity, making use of their homing effect, and using them as carriers for drug transport.

It was shown that overexpression of the plasma extracellular vesicular circFoxp1 could promote OC cell proliferation and reduce cisplatin sensitivity, while knockdown of circFoxp1 inhibited OC cell proliferation and enhanced chemotherapy effects [46]. In addition, extracellular vesicular miR21 conferred chemoresistance and an aggressive phenotype in OC cells through its transfer from neighboring stromal cells, indicating that preventing the extracellular vesicular transfer of miR21 from stromal cells could be a new strategy for suppressing OC growth [40]. In addition, studies have shown that the macrophage miR-7 is transferred to EOC cells by EVs and inhibits the EGFR/AKT/ERK1/2 signaling pathway, thereby inhibiting OC metastasis and invasiveness [41]. The transmembrane family protein 205 was considered to contribute to cellular platinum resistance via increased extracellular vesicular efflux of platinum agents, and one study used the transmembrane family protein 205 inhibitor L-2663 to selectively reduce the secretion of EVs and block the platinum efflux from platinum-resistant OC cells as a target for combination therapy [118]. Researchers isolated EVs from fresh milk, wrapped the chemotherapeutic drug cisplatin, and introduced the drug into cisplatin-resistant OC, which allowed the drug to escape endosomal capture and improved the anticancer effect of drug delivery on the cisplatin-resistant OC. Compared with the pure cisplatin treatment, it achieved a better therapeutic effect [23]. The OC cells’ extracellular vesicular miR-155-5p prevented the formation of an immunosuppressive tumor microenvironment by downregulating PD-L1 and other immunosuppressive factors, thus inhibiting OC development and macrophage infiltration [42]. Similarly, by downregulating the tumor-associated macrophages (TAMs)-derived EVs miR-29a-3, PD-L1 expression and OC cell proliferation and immune escape were inhibited [43].

As carriers of drug transport, EVs have low immunogenicity and toxicity, high circulating and tissue stability, and inherent homing ability [119]. One study used tumor-derived EVs to encapsulate miR497 to achieve targeted therapy and reduce drug resistance [44]. Another study improved vascular normalization by targeting miR-484 with RGD-modified EVs, thereby enhancing the OC chemosensitivity [24].

The pre-metastatic ecotone is a microenvironment formed by EVs secreted by tumors before extensive metastasis. The therapeutic effect of OC can be achieved using the homing ability of exosomes on primary cancer cells [120]. Researchers embedded OC ascites-derived EVs into 3D stents and implanted them in animal models to mimic pre-metastatic ecological sites. As a preferential site for cancer cell metastasis, promoting tumor cell adhesion in a nonpharmacological mode of action can prolong OC survival [121].

## 3. Progress in the Treatment of EVs in CC

CC is the fourth most common cancer in women after breast, colorectal, and lung cancer, and is one of the leading causes of female death worldwide, with most deaths occurring in low- and middle-income countries [2,122]. There are two common histological subtypes of CC: cervical squamous cell carcinoma and cervical adenoma [123]. The FIGO staging, which combines physical examination, endoscopic surgery, and imaging, is the most commonly used staging of CC [124]. High-risk human papillomavirus (HPV) is a common virus, and most people will be infected with it at some point in their lives [125]. Numerous studies have confirmed that persistent HPV infection is a necessary etiology for the formation of CC [126,127,128]. Although the detection and prevention of CC have made great progress, the survival rate of patients has not changed significantly [129]. Studies have demonstrated that EVs are associated with HPV infection and can play an essential role in different stages of CC development by promoting cell proliferation, mediating immune escape, and remodeling the tumor microenvironment. Therefore, they have become important biomarkers for CC diagnosis and treatment [65,69].

### 3.1. Development and Metastasis of CC

Zhang et al. found that the cervical squamous carcinoma cell-derived EVs miR-223 could target and transform the 3’-UTR of transforming growth factor beta receptor 3 and 3-hydroxy-3-methylglutaryl-coenzyme A synthase 1, inhibiting their expression, and promoting tumor growth. They also induced monocytes/macrophages to secrete IL-6, and the secretion of IL-6 activates the signal transducer and activator of transcription3 in cervical squamous cancer cells, expressing more miR-233, and accelerating tumor growth [57].

The role of EVs in CC angiogenesis has also been reported. The extracellular vesicular miR-221-3p secreted by CC cells promoted the invasion, migration, and angiogenesis of CC microvascular endothelial cells through downregulation of MAPK10 and THBS2 expression [58,59]. CC adenocyte-derived EVs were found to deliver tyrosine kinase with immunoglobulin and epidermal growth factor homology domains 2 (TIE2) to macrophages and promote tumor angiogenesis [52]. EVs from CC cells, especially from high-risk HPV-positive cells, promoted angiogenic responses in neighboring endothelial cells through the upregulation of the Hedgehog–GLI signaling pathway [130]. Cervical cancer-derived EVs were found to deliver miR-663b to human umbilical vein endothelial cells (HUVEC) and inhibit vinculin expression, thereby promoting vascular growth and tumor growth [61]. Other research showed that the extracellular vesicular miR-221-3p secreted by CC cells promoted lymphangiogenesis and lymphatic metastasis through the downregulation of vasohibin-1 signaling [60].

EVs from human HPV-positive cell lines were more effective in promoting neurite growth than EVs from HPV-negative cell lines. Further studies revealed that EVs derived from HPV-positive CC cell lines effectively stimulated neurite growth and mediated tumor innervation, affecting the tumor microenvironment [131]. The CC cell-derived extracellular vesicular miRNAs, including let-7d-5p, miR-20a-5p, miR-378a-3p, miR-423-3p, miR-7-5p, miR-92a-3p, and miR-21-5p, were regulated by viral E6/E7 oncogenes in HPV-positive tumor cells, thus affecting the growth of HPV-positive cancer cells [132]. The CC cell-derived extracellular vesicular Wnt2B, a member of the Wnt protein family, promoted the transformation of normal fibroblasts to CAFs by activating the Wnt/β-catenin signaling pathway, thereby promoting the growth and development of primary tumors [53]. The cancer cell-derived extracellular vesicular miR-663b was endocytosed by CC cells after TGF-β1 exposure and inhibited the expression of mannoside acetylglucosaminyltransferase 3, thus accelerating the epithelial–mesenchymal transition (EMT) process and ultimately promoting local and distant metastasis [62]. HIV-infected patients face a higher risk of HPV infection and CC [122]. It has been shown that T cell-derived extracellular vesicular miR-155-5p infected with HIV promoted the proliferation, migration, and invasion of CC cells [63]. The long noncoding RNA (lncRNA) LINC01305 was mainly distributed in EVs and was transferred to recipient cells to enhance CC progression [76]. The lncRNA AGAP2-AS1 from EVs was involved in CC cell proliferation by regulating the miR-3064-5P/SIRT1 signaling pathway [77]. EVs secreted by CSCC cells were found to deliver miR-142-5p to lymphatic endothelial cells via the ARID2-DNMT1 -IFN-γ signaling pathway, inducing the expression of indoleamine 2,3-dioxygenase, thereby inhibiting CD8^+^ T cells [64].

### 3.2. Diagnosis and Prognosis of CC

CC cell-derived EVs can pass through the interstitial cells and tissue fluids, thus collecting in body fluids such as cervical vaginal lavage and blood. They are potential markers for the diagnosis and prognosis of CC because they contain characteristic substances that reflect the physiological state of CC cells.

One study sequenced 121 plasma samples from healthy volunteers, CC patients, and cervical intraepithelial neoplasia patients for extracellular vesicular miRNA. The study found that plasma extracellular vesicular let-7d-3p and miR-30d-5p were differentially expressed in plasma samples and normal tissues adjacent to the tumor. They are potential biomarkers for the noninvasive screening of CC and precancerous lesions [65]. Liu et al. collected cervical vaginal lavage fluid specimens and isolated EVs from 45 CC patients, 25 HPV-positive subjects, and 32 HPV-negative subjects. They found increasing levels of miRNA-21 and miRNA-146a expression in extracellular vesicular from CC patients [66]. Another study collected 90 cervical vaginal lavages (30 CC patients, 30 HPV-positive non-cancerous volunteers, and 30 HPV-negative non-cancerous volunteers) and the Hox transcript antisense intergenic RNA (HOTAIR), MALAT1, and maternally expressed gene 3 (MEG3) of extracellular vesicular lncRNAs were extracted for quantification. These three lncRNAs were found to be clustered in CC EVs, with significant differences in expression between CC patients and non-cancerous patients, and had the potential for the detection and diagnosis of CC [78]. One study measured the expression level of miRNA in the plasma EVs of 97 patients with CC and 87 normal controls, and it was found that miR-146a-5p, miR-151a-3p, and miR-2110 were upregulated in plasma EVs, which were potential biomarkers [67]. In another study, plasma from CC patients and healthy controls was sequenced for extracellular vesicular miRNA, and the miR-125a-5p expression levels were significantly lower in CC patients than in healthy controls, which could be considered as a diagnostic marker for CC [68].

Pauline et al. identified multiple nuclear transporter proteins in CC cell-derived EVs and found that the combination of KPNβ1, CRM1, KPNα2, CAS, RAN, IPO5, and TNPO1 as a biomarker group had the highest diagnostic power for CC [54]. One study analyzed the transcriptional profiles of CC EVs and found significant differences in the tumor-promoting content and enrichment of mRNA from protumor cells and HPV E6, and these transcripts may serve as potential exosome biomarkers for CC [55]. Some scholars screened plasma extracellular vesicular RNAs from CC patients before and during cisplatin-based concurrent chemoradiotherapy). They found that miRNA (miR-142-3p), mRNAs (CXCL5, KIF2A, RGS18, APL6IP5, and DAPP1), and snoRNAs (SNORD17, SCARNA12, SNORA6, SNORA12, SCRNA1, SNORD97, SNORD62, and SNORD38A) in combination could clearly distinguish between normal and tumor specimens and could be used for the diagnosis of CC [75]. They also compared plasma extracellular vesicular miRNA before and during cisplatin-based concurrent chemoradiotherapy similarly and found a significant correlation between plasma extracellular vesicular miRNA and the degree of early tumor progression and metastasis [133].

Some scholars researched the serum extracellular vesicular lncRNA DLX6-AS1 levels in 111 patients with CC, 60 patients with cervical intraepithelial neoplasia, and 110 healthy women; it was found that the lncRNA DLX6-AS1 levels were significantly elevated in patients with CC. In addition, high expression of lncRNA DLX6-AS1 was positively correlated with lymph node metastasis, differentiation, FIGO staging, and shorter survival. Patients with a high expression of lncRNA DLX6-AS1 were prone to recurrence of CC. Therefore, serum extracellular vesicular lncRNA DLX6-AS1 is a potential marker for predicting the overall survival of CC patients [79]. One study reported that the CC cell-derived extracellular vesicular miR-1468-5p was highly expressed in peripheral serum blood and positively correlated with PD-L1^+^ lymphatic vessels and PD-1^+^ CD8^+^ T cells, thus being valuable in determining the prognosis of CC [69].

### 3.3. Treatment of CC

Exosomes are important in the proliferation, angiogenesis, and cell metastasis of CC, and are effective in the corresponding treatment. Jin et al. used 5-Aminolevulinic acid photodynamic therapy for the treatment of CC and found that the therapy inhibited miR-34a expression and increased high mobility group box1 (HMGB1) EVs’ secretion, thereby inhibiting cell proliferation and promoting apoptosis [56]. Another study demonstrated that mifepristone inhibited the CC HeLa cell-derived extracellular vesicles by upregulating their ISG15 protein expression levels, thereby inhibiting CC metastasis [134]. Zhang et al. found that the EVs of HPV-16 E7-pulsed DCs inhibited the migration and M1 polarization of macrophages, thereby blocking CC progression [135]. Yan et al. found that the extracellular vesicle miR-423-3p could inhibit macrophage M2 polarization, thereby suppressing CC cell progression and tumor growth [70].

Sensitivity to radiotherapy tends to decrease in the later stages of tumor treatment. Taking EVs as the target, some scholars have addressed the problem by inhibiting or increasing the release of EVs. Raji et al. found that miR-106a/b in cisplatin-resistant Hepatocarcinoma cells exosomes could reduce the resistance of CC cells to cisplatin by upregulating the level of silent information regulator 1 (SIRT1) in the CC cells [71]. Fang et al. found that miR-1323 was transferred by EVs derived from CAFs. Downregulation of miR-1323 inhibited cell proliferation, migration, invasion, and increased cell radiosensitivity in CC [72]. Konishi et al. found that miR-22-enriched EVs could alter the expression of the c-Myc binding protein and human telomerase reverse transcriptase genes in CC cells, thereby improving the sensitivity of in vitro CC radiotherapy [73]. Zhou et al. found that miR-320a expression was low in cisplatin-resistant CC. Myeloid Cell Leukemia Sequence 1, a cisplatin-resistant molecule, was regulated by engineered miR-320a EVs, thereby reducing resistance to cisplatin therapy in CC [74].

EVs can also act as drug carriers to enhance drug targeting and biocompatibility. Aqil et al. used milk-derived EVs as nanodrug carriers to compare the efficacy of the direct oral administration of curcumin with EV delivery for treating CC mice. They found EVs as carriers achieved better efficacy, and there were no significant biocompatibility issues [136].

## 4. Progress in the Treatment of EVs in EC

EC is the sixth most common cancer among women [2]. Over the past 30 years, the number of cases has increased by 132% [137]. Aging, obesity, polycystic ovary syndrome, and Lynch syndrome are common risk factors for EC [138]. Tumor metastasis is the main cause of death caused by EC. Most patients with early diagnosis of EC have a good prognosis, while about 20% of patients with local metastases and 10% of patients with distant metastases have poor survival results [139]. Over the past few decades, the treatment of EC has hardly changed, with surgery remaining the preferred modality. Adjuvant therapies include adjuvant radiotherapy, chemotherapy, and hormonal therapy, and there are still few treatments for EC metastases [139,140]. Therefore, diagnosis and treatment in the early stages of EC are effective ways to reduce EC mortality.

### 4.1. Development and Metastasis of EC

There is evidence that EVs are associated with the angiogenesis, growth, and development of EC cells and the metastatic potential of EC [141].

Song et al. found that plasma EVs from EC patients contained lectin galactoside-binding soluble 3-binding protein (LGALS3BP), which promoted tumor growth and angiogenesis through the PI3K/Akt/VEGFA signaling pathway [80]. Li’s study found that CAFs-derived EVs had reduced miR-148b expression compared to NFs, thereby increasing EC invasiveness [83]. Maida et al. found that EC cells delivered small regulatory RNAs to endometrial fibroblasts via EVs, and that EC cells may alter the peripheral stroma by transferring extracellular vesicular RNA [84]. Shi et al. found that miR-133a, which may regulate the downregulation of FOXL2 in EC tissues, was present in EC cell-derived EVs and could be delivered to normal endometrial cells [85]. Jia et al. found that EC cell-derived EVs deleted in lymphocytic leukemia 1 (DLEU1) accelerated EC progression by regulating the miR-381-3p/E2F3 pathway [98]. Another study showed that serum extracellular vesicular miR-27a-5p in patients with polycystic ovary syndrome enhanced the migration of EC cells, which led patients with polycystic ovary syndrome to EC progression [86]. Fan et al. found that the lncRNA nuclear-enriched abundant transcript 1 (NEAT1) existed in the EVs of CAFs, which interfered with the microenvironment of EC through the miR-26a/b-5p-STAT3-YKL-40 signaling pathway and promoted the development of EC [99]. Xiao et al. found that EC cells under hypoxic conditions promoted M2-like macrophage polarization through the delivery of extracellular vesicular miRNA-21, which altered the tumor immune microenvironment and could be a potential mechanism for promoting EC progression [87].

### 4.2. Diagnosis and Prognosis of EC

The current clinical screening of EC relies on a vaginal ultrasound and endometrial tissue biopsy, but both of them lack specificity [142]. Therefore, a method that can screen EC early and accurately is needed. Several studies analyzed biological fluids from EC patients and identified extracellular vesicular biomarkers with diagnostic and prognostic values.

Herrero et al. found that the Annexin 2 (ANXA2) protein was elevated in the plasma EVs of EC patients with specificity and sensitivity. Moreover, the ANXA2 levels were positively correlated with the risk of recurrence, demonstrating that ANXA2 was a potential diagnostic and prognostic biomarker [81]. Song et al. detected Serpin family A member 5 (SERPINA5) protein levels in plasma EVs. It was found that circulating plasma levels of the extracellular vesicles of SERPINA5 were elevated in EC patients, SERPINA5 expression was reduced in EC patients with distant metastases, and low SERPINA5 expression indicated poor survival [82]. Zhou et al. compared plasma from healthy subjects and EC patients and found that the extracellular vesicles of miR-15a-5p, miR-106b-5p, and miR107 were significantly upregulated in the plasma of EC patients compared to healthy subjects. In particular, the expression of the extracellular vesicles of miR-15a-5p was associated not only with the depth of the infiltration and invasiveness of EC, but also with the reproductive levels of testosterone and dehydroepiandrosterone sulfate, which is promising for the early diagnosis of EC [88]. Srivastava et al. compared urine-derived extracellular vesicular miRNA expression profiles from patients with EC and patients with symptoms of EC but no diagnosis and found that 54 of the 84 miRNAs studied were amplified in the qPCR, with hsa-miR-200c-3p as the most enriched, suggesting it as a biomarker for the diagnosis of EC [89]. Zheng et al. isolated EVs from the sera of 100 EC cases and 100 healthy control patients and extracted RNA from them. They found that increased extracellular vesicles of miRNA-95 and decreased miRNA-205 were associated with reduced overall survival in EC patients and may be prognostic biomarkers for EC patients [92]. Wang et al. sequenced plasma EVs from EC patients and normal humans. They found that the plasma-derived extracellular vesicular miR-26a-5p levels were significantly lower in EC patients, especially in patients with combined lymph node metastasis. Human lymphatic endothelial cells took up the extracellular vesicular miR-26a-5p secreted by EC cells, which induced lymphatic vessel formation by activating Lymphoid enhancer-binding factor 1. This can be used as a biomarker for early identification of lymph node metastasis risk in EC patients [90]. Fan et al. analyzed plasma miRNA from 93 patients with EC and 79 normal subjects. The miR-142-3p, miR-146a-5p, and miR-151a-5p were significantly elevated in the plasma of EC patients, and miR-151a-5p expression was also considerably elevated in EVs, showing a potential for the noninvasive diagnosis of EC [91].

### 4.3. Treatment of EC

Exosomes play an essential role in the development and progression of EC and provide new ideas for the treatment of EC. Song et al. found that overexpression of the SERPINA5 protein in plasma EVs could inhibit the metastatic potential of EC cells in vivo by inhibiting the integrin β1/FAK signaling pathway. In addition, the extracellular vesicular SERPINA5 protein hindered tumor growth and metastasis in xenograft models. Thus, the SERPINA5 EVs may be a new strategy for treating metastatic EC [82]. Wang et al. found that upregulation of miR-192-5p by TAMs-derived EVs inhibited the IRAK1/NF-kB signaling pathway, effectively promoting apoptosis and impeding EMT in EC cells, thereby inhibiting EC progression. The results proved that the miR-192-5P-modified TAMs-derived EVs may be a potential target for EC therapy [93]. Mojtahedin et al. used LED irradiation to alter exosome ontogenesis, angiogenic capacity, and metastatic behavior to produce tumor-suppressive effects on Ishikawa cells [143]. Gu et al. found that TAMs-derived EVs acted as carriers of hsa_circ_0001610, transferring hsa_circ_0001610 to EC cells and releasing cyclin B1 expression through the adsorption of miR-139-5p, thereby reducing the radiosensitivity of EC cells. This suggests that hsa_circ_0001610 could serve as a potential intervention for EC radioresistance [100]. Park et al. used Aurea helianthus extract to induce extracellular vesicular miRNAs in EC cells to reach the goal of prevention and treatment of EC in five aspects: inhibiting migration and invasion, increasing drug sensitivity, reducing inflammation, and promoting cellular senescence [144]. Zhou et al. showed in vivo and in vitro studies where CD45RO-CD8^+^ T cell-derived EVs released high levels of miR-765 and modulated the miR-765/PLP2 signaling pathway to inhibit the pro-tumorigenic effects of estrogen on uterine corpus EC. Zhang et al. found that the miR-320a EVs secreted by CAFs inhibited the proliferation of Ishikawa and HEC-1B cells by downregulating the hypoxia-inducible factor 1alpha/VEGFA signaling pathway, thereby inhibiting EC proliferation and improving the sensitivity of radiotherapy [94].

More scholars exploited the tumor homing effect of human umbilical cord mesenchymal stem cells (hUCMSCs) to explore the treatment of EC. Pan et al. investigated hUCMSCs-derived extracellular vesicular miRNA-503-3p and found that upregulation of EVs could inhibit mesoderm-specific transcript, thereby inhibiting EC growth [95]. Li et al. found that miR-302a delivered by hUCMSC-derived EV significantly reduced EC cell proliferation and migration by reducing the expression of AKT, reducing the phosphorylation of AKT, and downregulating the cyclin D1 expression, which may be an effective treatment for EV [96]. Liang et al. found that hUCMSCs-derived EVs delivered miR-499a-5p as a vector, which could be effectively taken up by Ishikawa cells and upregulate miR-499a-5p expression. miR-499a-5p EVs inhibited the proliferation of Ishikawa and HUVEC cells through VAV3 gene targeting. Moreover, they inhibited the pro-angiogenic ability of HUVECs cells, thus inhibiting tumor growth and angiogenesis in EC and exerting tumor suppressive effects [97].

## 5. Conclusions

Gynecological malignancies, especially OC, CC, and EC, cause high economic burdens and physical damage to patients across the globe. With the progress in the medical field, the diagnosis and treatment of gynecological malignant tumors are constantly developing and improving. EVs widely exist in living organisms, whose biological functions are gradually being recognized, and research involving their role in diagnosis and their therapeutic applications is being carried out. Understanding how EVs serve as biological markers, tumor-promoting factors, tumor-suppressing factors, and targeting vectors in the development of gynecologic tumor diseases can help to explain their biological functions more comprehensively and objectively and provide clinical assistance in the diagnosis and treatment of gynecologic tumors.

Although vesicle diagnosis and treatment techniques have shown great diagnostic and therapeutic potential for gynecologic tumors, research in the field is still at the exploratory stage. In addition, the immaturity of the research methods hinders its translation into clinical application. A deeper understanding of the formation and regulatory mechanisms of EVs will facilitate a better understanding of the biological functions, heterogeneity, and functional diversity of EVs, as well as efficiently prepare and improve unified and standardized EVs, therefore better realizing their clinical significance.

## Figures and Tables

**Table 1 bioengineering-09-00740-t001:** Role of EV biomarkers in the gynecologic cancer microenvironment.

Marker Type	Cancer Type	EV Marker	EV Source	Function	Potential Clinical Application	Ref.
Protein	OC	ATF2	Cells	Enhance angiogenesis	Therapeutic target	[16]
		MTA1				
		ROCK1/2				
		sE-cad	Serum, ascites	Enhance angiogenesis		[17]
		claudin-4	Cells	Biomarker	Early detection	[18]
		HGF	Serum	Biomarker		[19]
		STAT3		Biomarker		
		IL-6		Biomarker		
		TGFβ	Cells	Biomarker		[20]
		FGF9	Cells	Biomarker	Prognosis prediction	[21]
		FATS	Plasma	Biomarker		[22]
		Clathrin	Milk	Enhance the anti-cancer effectiveness of cisplatin	Nanocarrier	[23]
		LAMP2B	Cells	Enhance the sensitivity of chemotherapy		[24]
Protein receptor		PKR1	Serum	Enhance angiogenesis	Therapeutic target	[25]
miRNA		miR-130a	Cells	Enhance angiogenesis		[26]
		miR-205	Cells	Enhance angiogenesis		[27]
		miR-141-3p	Cells	Enhance angiogenesis		[28]
		miR-940	Ascites	Stimulate TAM polarization		[29]
		miR-221-3p	Serum	Promote cancer cells invasion and migration		[30]
		miR-6780b-5p	Ascites	Promote cancer cells invasion and migration		[31]
		miR-21-5p	Cells, Plasma	Promote cancer cells invasion and migration		[32]
		miR-21	Serum	Biomarker	Early detection	[33]
		miR-141	Serum	Biomarker		
		miR-200a	Serum	Biomarker		
		miR-200b	Serum	Biomarker		
		miR-200c	Serum	Biomarker		
		miR-203	Serum	Biomarker		
		miR-205	Serum	Biomarker		
		miR-214	Serum	Biomarker		
		miR-21	Plasma	Biomarker		[34]
		miR-100	Plasma	Biomarker		
		miR-200b	Plasma	Biomarker		
miRNA	OC	miR-320	Plasma	Biomarker	Early detection	[34]
		miR-16	Plasma	Biomarker		
		miR-93	Plasma	Biomarker		
		miR-126	Plasma	Biomarker		
		miR-223	Plasma	Biomarker		
		miR-1290	Serum	Biomarker		[35]
		miR-1260a	Plasma	Biomarker		[36]
		miR-7977	Plasma	Biomarker		
		miR-192-5p	Plasma	Biomarker		
		miR-21-5p	Cells	Biomarker		[37]
		miR-29a-3p	Cells	Biomarker		
		miR-200a	Ascites	Biomarker	Prognosis prediction	[38]
		miR-200b	Ascites	Biomarker		
		miR-200c	Ascites	Biomarker		
		miR-1290	Ascites	Biomarker		
		miR-484	Serum	Biomarker		[39]
		miR21	Cells	Increase chemoresistance	Therapeutic target	[40]
		miR-7	Cells	Inhibit cancer cells invasion and migration		[41]
		miR-155-5p	Cells	Inhibit cancer cells invasion and migration		[42]
		miR-29a-3p	Cells	Promote cancer cells invasion and migration		[43]
		miR497	Cells	Reduce cisplatin resistance	Nanocarrier	[44]
circRNA		circRNA051239	Plasma	Promote cancer cells invasion and migration	Therapeutic target	[45]
		circFoxp1	Serum	Biomarker and increase cisplatin resistance	Prognosis prediction and therapeutic target	[46]
		Cdr1as	Serum	Biomarker	Prognosis prediction	[47]
LncRNA		MALAT1	Serum	Enhance angiogenesis and biomarker	Therapeutic target and prognosis prediction	[48]
		lncRNA ATB	Cells	Enhance angiogenesis	Therapeutic target	[49]
Lipid		PS	Ascites	Inhibit T cell activation		[50]
Enzyme		ARG-1	Plasma, ascites	Inhibit T cell proliferation		[51]
Protein	CC	TIE2	Cells	Enhance angiogenesis		[52]
		Wnt2B	Cells	Enhance stroma remodeling and cancer progression		[53]
		KPNβ1	Serum	Biomarker	Early detection	[54]
		CRM1				
		CAS				
		IPO5				
		TNPO1				
		HPV E6	Cells	Biomarker	Early detection	[55]
		HMGB1	Cells	Inhibit cell proliferation and promote cell apoptosis	Therapeutic target	[56]
miRNA		miR-223	Cells	Activates STAT3 signals		[57]
		miR-221-3p	Cells	Enhance angiogenesis		[58,59]
			Cells	Enhance lymphangiogenesis and lymphatic metastasis		[60]
		miR-663b	Cells	Enhance angiogenesis		[61]
			Cells	Enhance the metastatic ability of cancer cells		[62]
		miR-155-5p	Cells	Promote cancer cells invasion and migration		[63]
		miR-142-5p	Cells	Suppress and exhaust CD8 T cells		[64]
		Let-7d-3p	Plasma	Biomarker	Early detection	[65]
		miR-30d-5p				
		miR-21	Cervicovaginal lavage sample	Biomarker		[66]
		miR-146a				
		miR-146a-5p	Plasma	Biomarker		[67]
		miR-151a-3p				
		miR-2110				
		miR-21-5p				
		miR-125a-5p	Plasma	Biomarker		[68]
		miR-1468-5p	Cells	Biomarker	Prognosis prediction	[69]
miRNA	CC	miR-423-3p	Plasma	Inhibit the macrophage M2 polarization	Therapeutic target	[70]
		miR-106a	Cells	Reduce cisplatin resistance		[71]
		miR-106b				
		miR-1323	Cells	Enhance cancer progression and radioresistance		[72]
		miR-22	Cells	Enhance the sensitivity of radiotherapy		[73]
		miR-320a	Cells	Reduce cisplatin resistance	Nanocarrier	[74]
		miR-142-3p	Plasma	Biomarker	Early detection	[75]
mRNA		CXCL5				
		KIF2A				
		RGS18				
		APL6IP5				
		DAPP1				
snoRNA		SNORD17				
		SCARNA12				
		SNORA6				
		SNORA12				
		SCRNA1				
		SNORD97				
		SNORD62				
		SNORD38A				
lncRNA		LINC01305	Cells	Enhance cancer progression	Therapeutic target	[76]
		AGAP2-AS1	Cells	Promote cancer cells invasion and migration		[77]
		HOTAIR	Cervicovaginal lavage sample	Biomarker	Early detection	[78]
		MALAT1				
		MEG3				
		DLX6-AS1	Serum	Biomarker	Prognosis prediction	[79]
Protein	EC	LGALS3BP	Plasma	Promote cancer cells invasion and migration, enhance angiogenesis	Therapeutic target	[80]
		ANXA2	Plasma	Biomarker	Early detection	[81]
		SERPINA5	Plasma	Biomarker	Prognosis prediction	[82]
miRNA		miR-148b	Cells	Inhibit cancer cells invasion and migration	Therapeutic target	[83]
		miR-141-3p	Cells	Intercellular communication between cancer cells and neighboring fibroblasts		[84]
		miR-200b-3p	Cells			
		miR-133a	Cells	Enhance cancer progression		[85]
		miR-27a-5p	Serum	Promote cancer cells invasion and migration		[86]
		miRNA-21	Cells	Enhance the macrophage M2 polarization		[87]
		miR-15a-5p	Plasma	Biomarker	Early detection	[88]
		miR-200c	Urine	Biomarker		[89]
		miR-26a-5p	Plasma	Biomarker		[90]
		miR-142-3p	Plasma	Biomarker		[91]
		miR-146a-5p				
		miR-151a-5p				
		miRNA-93	Serum	Biomarker	Prognosis prediction	[92]
		miRNA-205				
		miR-192-5p	Cells	Inhibit cancer cells EMT and metastasis	Therapeutic target	[93]
		miR-320a	Cells	Inhibit cancer cells invasion and migration		[94]
		miR-503-3p	Cells	Inhibit cancer cells invasion and migration		[95]
		miR-302a	Cells	Inhibit cancer cells invasion and migration		[96]
		miR-499a-5p	Cells	Inhibit cancer cells invasion and migration, inhibit angiogenesis		[97]
lncRNA		DLEU1	Cells	Promote cancer cells invasion and migration		[98]
		NEAT1				[99]
circRNA		hsa_circ_0001610	Cells	Reduce the sensitivity of radiotherapy		[100]

## Data Availability

No new data were created or analyzed in this study. Data sharing is not applicable to this article.

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
