# Peer review of "Application of Extracellular Vesicles in Gynecologic Cancer Treatment"

_bioengineering, 2022, doi:10.3390/bioengineering9120740_

Round 1

Reviewer 1 Report

-The review summarizes the potential involvement of extracellular vesicles in diagnosis and treatments of gynecological cancers.

-The writing is very poor and full of typos and grammatical errors. It is very hard to understand some sentences. Extensive editing of English writing is required.

-Moreover, the authors are advised to read the recent articles and classify the vesicles according to MISEV2018 report. With a special attention to the size of exosomes and the molecular identification and characterization.

-In many places, and in the title, authors mention "vesicles" alone, please correct it into "extracellular vesicles".

-Since the title is for "drug delivery", so authors are advised to add a subheading regarding the different methods used for loading the extracellular vesicles with the drugs, nucleic acids (mRNA and miRNA), and others.

-Figure 1 doesn't add any value, authors are advised to change it with focusing on the gynecologic cancers (diagnosis and therapeutic targets).

Author Response

Reviewer 1

The review summarizes the potential involvement of extracellular vesicles in diagnosis and treatments of gynecological cancers.

Comment 1: The writing is very poor and full of typos and grammatical errors. It is very hard to understand some sentences. Extensive editing of English writing is required.

Response: We apologize for the poor language of our manuscript. We have now worked on both language and readability and have also involved native English speakers for language corrections. We really hope that the flow and language level have been substantially improved.

Comment 2: Moreover, the authors are advised to read the recent articles and classify the vesicles according to MISEV2018 report. With a special attention to the size of exosomes and the molecular identification and characterization.

Response: We followed the latest ISEV guidelines and corrected the description of the classification of extracellular vesicles in this manuscript.

Comment 3: In many places, and in the title, authors mention "vesicles" alone, please correct it into "extracellular vesicles".

Response: Thank you for the suggestion. The description has been corrected.

Comment 4: Since the title is for "drug delivery", so authors are advised to add a subheading regarding the different methods used for loading the extracellular vesicles with the drugs, nucleic acids (mRNA and miRNA), and others.

Response: Thanks for the reviewer’s suggestion. We propose as a new title: “Application of extracellular vesicles in gynecologic cancer treatment”. Focus on the overview of extracellular vesicles for gynecologic tumor therapy rather than solely delivery functions.

Comment 5: Figure 1 doesn't add any value, authors are advised to change it with focusing on the gynecologic cancers (diagnosis and therapeutic targets).

Response: Thanks for the reviewer’s suggestion. We have deleted Figure 1 and added Table 1 to summarize the application of EVs in gynecological cancers.

Reviewer 2 Report

The authors of the work "application of vesicle-based delivery systems for gynecologic cancer treatment" make an updated review of the role of membrane vesicles in ovarian cancer, cervical cancer and endometrial cancer. I have to make a number of comments.

- First of all, I would replace the expression “more and more” with a synonym. It can be found in both line number 13 and 31, for example, and could be replaced by "several". Likewise, the commas before an "and" should also be removed.

- In line 27 I miss a reference that collects information on the presence of EVs in different body fluids.

- Regarding the size and types of EVs (lines 27-29) and figure 1, the sizes indicated for exosomes do not coincide in text and figure: 30-150 nm in text, 300-450 nm in figure.

- Throughout the text words appear in capital letters in the middle of the sentences. For example, “Microvesicles (MV)” on line 27, “Exosomes” on line 190, or “Bioinducible” on line 192. Please correct.

- “Exosomes can be detected in most body fluids” (line 41) would already be said (lines 26 and 27), so it should be removed.

- However, I think that to better justify why PVCs should be studied, part of the final paragraph of the introduction should be moved to the beginning of the introduction (“these cancers are important, with such statistics and, they produce PVCs, which are this and have the following characteristics”).

- The authors name the exosomes as miR-XXX-XX (“The exosomes let-7d-5p, miR-20a-5p (…) secreted by cervical cancer cells are regulated (…)”, lines 253-254, for example) . I am not aware of a nomenclature and classification system for exosomes, could the authors indicate it? As it is described, you might think that they are talking about miRNAs or even proteins (line 81).

- Beyond the use, detection or study of the EVs associated with the three types of cancer exposed, the authors also address, for each of the cancers, their development and metastasis, as well as their diagnosis and treatment. To give greater emphasis to the role of EVs, a summary table should be made specifically for the role of EVs based on the type of cancer and demonstrated effect.

- In section 3.3. “treatment” is written in lowercase (line 327). Please, correct.

Author Response

Reviewer 2

The authors of the work "application of vesicle-based delivery systems for gynecologic cancer treatment" make an updated review of the role of membrane vesicles in ovarian cancer, cervical cancer and endometrial cancer. I have to make a number of comments.

Comment 1: First of all, I would replace the expression “more and more” with a synonym. It can be found in both line number 13 and 31, for example, and could be replaced by "several". Likewise, the commas before an "and" should also be removed.

Response: We apologize for the poor language of our manuscript. We have now worked on both language and readability and have also involved native English speakers for language corrections. We really hope that the flow and language level have been substantially improved.

Comment 2: In line 27 I miss a reference that collects information on the presence of EVs in different body fluids.

Response: Thank you for this suggestion. We have added the missing references.

Comment 3: Regarding the size and types of EVs (lines 27-29) and figure 1, the sizes indicated for exosomes do not coincide in text and figure: 30-150 nm in text, 300-450 nm in figure.

Response: We followed the latest ISEV guidelines and corrected the description of the classification of extracellular vesicles in this manuscript.

Comment 4: Throughout the text words appear in capital letters in the middle of the sentences. For example, “Microvesicles (MV)” on line 27, “Exosomes” on line 190, or “Bioinducible” on line 192. Please correct.

Response: We are sorry for the incorrect expression and have corrected it.

Comment 5: “Exosomes can be detected in most body fluids” (line 41) would already be said (lines 26 and 27), so it should be removed.

Response: Thanks for your careful checking and kind reminder. This paragraph (line 38 and 41) should be a figure legend, and incorrectly placed to this location.

Comment 6: However, I think that to better justify why PVCs should be studied, part of the final paragraph of the introduction should be moved to the beginning of the introduction (“these cancers are important, with such statistics and, they produce PVCs, which are this and have the following characteristics”).

Response: We appreciate this valuable advice. We adapted the content of the introduction section to make it more logical.

Comment 7: The authors name the exosomes as miR-XXX-XX (“The exosomes let-7d-5p, miR-20a-5p (…) secreted by cervical cancer cells are regulated (…)”, lines 253-254, for example) . I am not aware of a nomenclature and classification system for exosomes, could the authors indicate it? As it is described, you might think that they are talking about miRNAs or even proteins (line 81).

Response: According to the International Society for Extracellular Vesicles (ISEV) nomenclature, researchers are suggested to consider using operational terms for EVs subtypes that refer to physical characteristics of EVs, biochemical composition and descriptions of conditions or cell of origin. This article also discusses about the role of the active components of EVs. The inaccurate description in the text has been corrected.

Comment 8: Beyond the use, detection or study of the EVs associated with the three types of cancer exposed, the authors also address, for each of the cancers, their development and metastasis, as well as their diagnosis and treatment. To give greater emphasis to the role of EVs, a summary table should be made specifically for the role of EVs based on the type of cancer and demonstrated effect.

Response: Thank you for the positive comments about the analyses. At your suggestion, we have now summarized the relevant information into several tables.

Comment 9: In section 3.3. “treatment” is written in lowercase (line 327). Please, correct.

Response: We are sorry for the incorrect expression and have corrected it.

Reviewer 3 Report

In their manuscript entitled “ Application of vesicle-based delivery systems for gynecologic cancer treatment”, Zhang et al present a review summarizing the main roles of extracellular vesicles (EVs) in gynecological cancers. While the manuscript offers a comprehensive overview of EVs content and impact on cancer progression, the title appears to be misleading. One would indeed expect from such a title a review describing new developments in the field of nanovectors, nanoencapsulation, targeted delivery… etc, but not a general focus on EVs and their potential as biomarkers with very little information describing EV-based delivery.

The manuscript would also greatly benefit from an extensive editing by a native English speaker. In its current state, too many sentences are either truncated or difficult to understand.

In many instances (some listed below), references are incorrect. Authors should carefully check all references, and correct when necessary.

Finally, most of the manuscript could be summarized in tables. Indeed, most of the text lists EV contents and their impact on cancer progression. Tables would help the readers to more easily access the relevant information, and would better illustrate the complexity and multiplicity of EV contents and effects.

Minor comments:

Lines 22-24: the description proposed by the authors is misleading. Vesicles are not usually described as being composed of 2 layers, one being the cytosol. Please amend the text.

Line 52: references 15 and 16 are not related to the sentence where they are placed.

Line 61: reference 18 is about colorectal cancer, and is therefore irrelevant to the present manuscript.

Line 94: “Ultimately, ovarian cancer metastasis.” Do the authors mean “metastasizes”?

Lines 152-153: “Conventional targeted drug-controlled systems for oncology treatment cause serious side effects, including organ toxicity and immune response”. This sentence is not clear: what happens to the immune response? It is usually downregulated during cancer treatment, but the current statement is ambiguous.

Line 169: CAF stands for Cancer-Associated Fibroblasts.

Line 215: CSCC stands for Cervical Squamous Cell Carcinoma

Line 216: Reference 79 is not linked to cervical cancer.

Line 219: Reference 80 describes the correlation between HIV infection and HPV-induced cervical cancer. It may not be the most apropos reference here.

Line 227: References 83 and 84 do not mention extracellular vesicles. Could the authors please explain why they chose these articles to illustrate that “they (EVs?) have become important biomarkers for cervical cancer diagnosis and treatment”?

Line 262: “MCAT3” should be replaced by MGAT3.

Line 291: MALAT1 stands for Metastasis-Associated Lung Adenocarcinoma Transcript 1.

Line 375: lectin instead of “electin”

Author Response

Reviewer 3

Comment 1: In their manuscript entitled “ Application of vesicle-based delivery systems for gynecologic cancer treatment”, Zhang et al present a review summarizing the main roles of extracellular vesicles (EVs) in gynecological cancers. While the manuscript offers a comprehensive overview of EVs content and impact on cancer progression, the title appears to be misleading. One would indeed expect from such a title a review describing new developments in the field of nanovectors, nanoencapsulation, targeted delivery… etc, but not a general focus on EVs and their potential as biomarkers with very little information describing EV-based delivery.

Response: We propose as a new title: “Application of extracellular vesicles in gynecologic cancer treatment”. Focus on the overview of extracellular vesicles for gynecologic tumor therapy rather than solely delivery functions.

Comment 2: The manuscript would also greatly benefit from an extensive editing by a native English speaker. In its current state, too many sentences are either truncated or difficult to understand.

Response: We apologize for the poor language of our manuscript. We have now worked on both language and readability and have also involved native English speakers for language corrections. We really hope that the flow and language level have been substantially improved.

Comment 3: In many instances (some listed below), references are incorrect. Authors should carefully check all references, and correct when necessary.

Response: We thank the reviewer for calling attention to this error, and have made the appropriate corrections.

Comment 4: Finally, most of the manuscript could be summarized in tables. Indeed, most of the text lists EV contents and their impact on cancer progression. Tables would help the readers to more easily access the relevant information, and would better illustrate the complexity and multiplicity of EV contents and effects.

Response: Thank you for the positive comments about the analyses. At your suggestion, we have now summarized the relevant information into several tables.

Minor comments:

Comment 5: Lines 22-24: the description proposed by the authors is misleading. Vesicles are not usually described as being composed of 2 layers, one being the cytosol. Please amend the text.

Response: We agree with the reviewers on the description of vesicles and delete the description of “double”.

Comment 6: Line 52: references 15 and 16 are not related to the sentence where they are placed.

Response: We thank the reviewer for calling attention to this error, and have made the appropriate corrections.

Comment 7: Line 61: reference 18 is about colorectal cancer, and is therefore irrelevant to the present manuscript. Line 227: References 83 and 84 do not mention extracellular vesicles. Could the authors please explain why they chose these articles to illustrate that “they (EVs?) have become important biomarkers for cervical cancer diagnosis and treatment”?

Response: We have proofread the whole manuscript and modified the inaccurate citation details and improper related statements accordingly.

Comment 8: Line 94: “Ultimately, ovarian cancer metastasis.” Do the authors mean “metastasizes”?

Response: We apologized for the confusion. We have modified the sentence to remove the ambiguity.

Comment 9: Lines 152-153: “Conventional targeted drug-controlled systems for oncology treatment cause serious side effects, including organ toxicity and immune response”. This sentence is not clear: what happens to the immune response? It is usually downregulated during cancer treatment, but the current statement is ambiguous.

Response: Thanks for the reviewer’s suggestion. We have modified the sentence to remove the ambiguity.

Comment 10: Line 169: CAF stands for Cancer-Associated Fibroblasts.

Response: Thanks for your careful checking and kind reminder. We have corrected the use of acronyms.

Comment 11: Line 215: CSCC stands for Cervical Squamous Cell Carcinoma.

Response: Thanks for your careful checking and kind reminder. We have corrected the use of acronyms.

Comment 12: Line 216: Reference 79 is not linked to cervical cancer.

Response: We thank the reviewer for calling attention to this error, and have made the appropriate corrections.

Comment 13: Line 219: Reference 80 describes the correlation between HIV infection and HPV-induced cervical cancer. It may not be the most apropos reference here.

Response: We thank the reviewer for calling attention to this error, and have added more relevant references.

Comment 14: Line 227: References 83 and 84 do not mention extracellular vesicles. Could the authors please explain why they chose these articles to illustrate that “they (EVs?) have become important biomarkers for cervical cancer diagnosis and treatment”?

Response: Thanks for your careful checking and kind reminder. We have replaced the corresponding references.

Comment 15: Line 262: “MCAT3” should be replaced by MGAT3.

Response: Thanks for your careful checking and kind reminder. We have corrected the use of acronyms.

Comment 16: Line 291: MALAT1 stands for Metastasis-Associated Lung Adenocarcinoma Transcript 1.

Response: Thanks for your careful checking and kind reminder. We have corrected the use of acronyms.

Comment 16: Line 375: lectin instead of “electin”.

Response: Thanks for your careful checking and kind reminder. We have corrected the use of acronyms.

Reviewer 4 Report

Zhang and colleagues propose a nicely written Review concerning advancements on vesicle-based delivery systems for gynecologic cancer treatment.

The manuscript is sounded and well written. I have two concerns:

1. In the whole manuscript authors write exosomes and microvesicles. Following the most recent ISEV guidelines these terms are old-fashioned and should be replaced with small- or large-vesicles. Please define in more details the different EVs subtypes and if exosomes or microvesicles terms are crucial for the manuscript, please clearly describe how they can fit the most recent nomenclature by ISEV.

2. The concept of EVs for the diagnosis is clearly described. Please add a paragraph about the techniques nowadays used for EVs detection in the analysis laboratory.

Author Response

Reviewer 4

Zhang and colleagues propose a nicely written Review concerning advancements on vesicle-based delivery systems for gynecologic cancer treatment.

The manuscript is sounded and well written. I have two concerns:

Comment 1: In the whole manuscript authors write exosomes and microvesicles. Following the most recent ISEV guidelines these terms are old-fashioned and should be replaced with small- or large-vesicles. Please define in more details the different EVs subtypes and if exosomes or microvesicles terms are crucial for the manuscript, please clearly describe how they can fit the most recent nomenclature by ISEV.

Response: Thank you for your valuable advice. We followed the latest ISEV guidelines and corrected the description of the classification of extracellular vesicles in this manuscript.

Comment 2: The concept of EVs for the diagnosis is clearly described. Please add a paragraph about the techniques nowadays used for EVs detection in the analysis laboratory.

Response: Thank you for the reviewer’s suggestion. We added Table 1 of Supplementary Materials to summarize the techniques of EVs isolation.

Round 2

Reviewer 1 Report

The manuscript has been improved but Table 1 needs more care. Authors may classify the markers type as (protein, mRNA, miRNA, and other nucleic acids) for easy interpretation. Moreover, write the full name of abbreviations below the table as a legend.

Author Response

Comment 1: The manuscript has been improved but Table 1 needs more care. Authors may classify the markers type as (protein, mRNA, miRNA, and other nucleic acids) for easy interpretation.

Response: We appreciate the reviewer for this valuable advice. We adjusted the arrangement of Table 1 based on markers type and cancer type to make it easier to understand.

Comment 2: Moreover, write the full name of abbreviations below the table as a legend.

Response: We thank the reviewer for this valuable advice. We checked the abbreviations in the article and removed unnecessary abbreviations. We have added a general table of abbreviations in the supplementary materials.

Reviewer 3 Report

The authors have addressed my issues.
I have no further comments.

Author Response

The authors have addressed my issues. I have no further comments.

Response: Thank you very much for your comments.